# Living on the edge: The sensitivity of arthropods to development and climate along an urban-wildland interface in the Sonoran Desert of central Arizona

Derek A. Uhey[1]*, Richard W. Hofstetter[1], Stevan Earl[2], Jerry Holden[3], Tiffany Sprague[4], Helen Rowe[3,5]

1 School of Forestry, Northern Arizona University, Flagstaff, Arizona, 2 Global Institute of Sustainability and Innovation, Arizona State University, Tempe, Arizona, 3 McDowell Sonoran Conservancy Citizen Science Program, Scottsdale, Arizona, 4 Arizona Game and Fish, Phoenix, Arizona, 5 School of Earth and Sustainability, Northern Arizona University, Flagstaff, Arizona

* Derek.Uhey@nau.edu

**Data Availability Statement:** Arthropod abundance data are available through the Environmental Data Initiative data repository.

## Abstract

Preservation of undeveloped land near urban areas is a common conservation practice. However, ecological processes may still be affected by adjacent anthropogenic activities. Ground-dwelling arthropods are a diverse group of organisms that are critical to ecological processes such as nutrient cycling, which are sensitive to anthropogenic activities. Here, we study arthropod dynamics in a preserve located in a heavily urbanized part of the Sonoran Desert, Arizona, U.S.. We compared arthropod biodiversity and community composition at ten locations, four paired sites representing the urban edge and one pair in the Preserve interior. In total, we captured and identified 25,477 arthropod individuals belonging to 287 lowest practical taxa (LPT) over eight years of sampling. This included 192 LPTs shared between interior and edge sites, with 44 LPTs occurring exclusively in interior sites and 48 LPTs occurring exclusively in edge sites. We found two site pairs had higher arthropod richness on the preserve interior, but results for evenness were mixed among site pairs. Compositionally, the interior and edge sites were more than 40% dissimilar, driven by species turnover. Importantly, we found that some differences were only apparent seasonally; for example edge sites had more fire ants than interior sites only during the summer. We also found that temperature and precipitation were strong predictors of arthropod composition. Our study highlights that climate can interact with urban edge effects on arthropod biodiversity.

## Introduction

Preserving undeveloped land near urban areas is a common strategy to maintain ecological integrity and processes of natural areas [1]. However, ecological processes do not recognize political boundaries and the extent to which ecosystem structure and function in protected

Relevant code and ancillary information are provided in Supporting Information files. Citation: Earl S, Grimm N, Childers D. Long-term monitoring of ground-dwelling arthropods in the McDowell Sonoran Preserve, Scottsdale, Arizona, ongoing since 2012 ver 4. Environmental Data Initiative. 2021. Available from: https://doi.org/10.6073/pasta/e596e3c6abbeaa0d1355dc3eb31f2c33 (Accessed 2023-09-07).

**Funding:** This material is based upon work supported by the National Science Foundation under grant number DEB-2224662, Central Arizona-Phoenix Long-Term Ecological Research Program (CAP LTER). The funders had no role in study design, data collection and analysis, decision to publish, or preparation of the manuscript.

**Competing interests:** The authors have declared that no competing interests exist.

areas are affected by anthropogenic activities in adjacent areas is not clear [2]. Urbanization alters biological communities, often replacing native species with exotics or eliminating them through habitat loss [3]. The effective management of protected areas depends on detailed knowledge of the biota and ecology, and monitoring ecological indicators [4].

Arthropods are well suited for monitoring ecological health [5]. Arthropods are diverse, abundant, and ecologically prominent, making their community composition indicative of ecological integrity [6]. Urbanization generally decreases native and/or specialist arthropods (e.g., [7, 8]), and may enhance nonnative and/or generalist arthropods (e.g., [9]). These effects can spill over through edge-effects onto arthropod communities in preserved areas bordering urbanized areas ([10–12]), but few studies have investigated these dynamics in arid regions.

Arthropods in arid regions face unique challenges and have evolved a range of adaptations to survive in hot and dry environments, such as the regulation of water loss and unique feeding habits [13]. These adaptations and general environmental stress make arthropod communities in arid regions vulnerable to disturbances such as urbanization. Urbanization in arid climates is associated with lower arthropod diversity and changes in community composition through the introduction of new species compared to undisturbed desert areas (e.g., [14]). Invasive species may dominate urbanized areas, resulting in lower community evenness [12]. Natural desert habitats support many native arthropod species, particularly those at high trophic levels that are not found in urbanized or fragmented habitats [15]. However, it is not known whether these urban pressures extend into the edge habitats of natural areas in arid environments.

Our study aimed to fill this knowledge gap by investigating arthropod community dynamics in an urban interface environment in the Sonoran Desert. The Sonoran Desert is an arid region with many areas fragmented or destroyed by rapid urbanization over the last seven decades [16]. This ecosystem is characterized by extreme seasonal variations in temperature and rainfall, with a high biodiversity. We examined arthropod communities of a large preserve in the Sonoran Desert near Scottsdale, Arizona, USA collected over eight years at paired locations at the urban edge and in the interior of the Preserve. We hypothesized that (1) habitats near the urban edge would have lower arthropod richness and abundance, and (2) habitats on the urban edge would have different arthropod compositions than habitats on the Preserve interior. We also investigated the role of climate–urbanization interactions by comparing seasonal and weather trends. Additionally, we hypothesized that (3) precipitation and temperature are strong drivers of arthropod community dynamics, and (4) changes to arthropod communities from the urban edge differ seasonally, as some seasons present more stressful climatic conditions. This research is important not only for understanding the dynamics of arthropod communities in arid regions but also for developing conservation strategies that may mitigate the negative impacts of urbanization on ecosystems.

## Materials and methods

### Study sites

We examined the arthropod communities at five locations (S1 Table, Fig 1) across the McDowell Sonoran Preserve in Scottsdale, Arizona, USA, established for long-term monitoring as part of the Central Arizona–Phoenix Long-Term Ecological Research (CAP LTER) program (https://sustainability-innovation.asu.edu/caplter/). The McDowell Sonoran Preserve, one of the world's largest urban preserves (~125 km$^2$), is adjacent to heavily urbanized parts of the greater Phoenix, Arizona metropolitan area. At four locations (Fig 1, S1 Table), we created paired sites with one set located within 100m of urban development (edge sites) and the other set located >0.5km away from the Preserve boundary (interior sites). We refer to proximity to the urban edge as treatment, with sites either being classified as interior or edge. We sampled

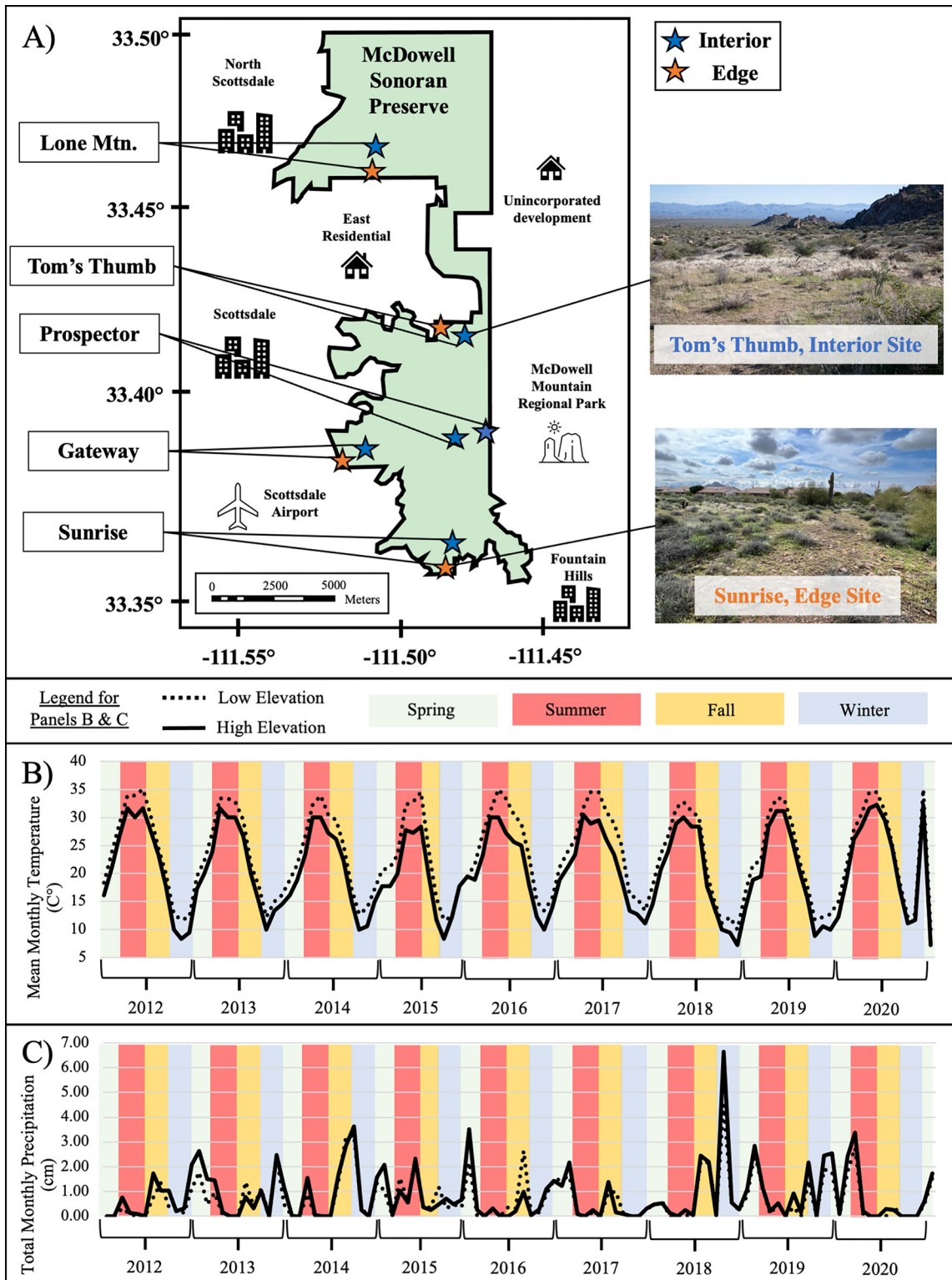

**Fig 1. Site locations and characteristics.** Location of paired sampling sites within the McDowell Sonoran Preserve, Scottsdale, Arizona and photos of two sites (panel A), mean monthly temperature over sampling period of low- (Gateway at 515m) and high-elevation sites (Tom's Thumb at 892m, Panel B), and total monthly precipitation over sampling period of low- and high-elevation sites (panel C). Sites span an elevational gradient from 515m to 892m. The Preserve boundary is outlined in black, with large portions bordering urban development. Edge sites (orange) are within 100m of the Preserve boundary whereas interior sites (blue) are at least 0.5km from the Preserve boundary.

**Table 1. Site soil and vegetation.** Soil type and plant community associations for each site. Soil types are from the Web Soil Survey and plant community associations are from Jones & Hull (2014).

| Location | Interior/ Edge | Soil Type | Plant Community |
|---|---|---|---|
| Lone Mountain | Interior | Clay Loam Upland | Mixed scrub- *Simmondsia chinensis*, *Larrea tridentata* |
| Lone Mountain | Edge | Clay Loam Upland | Mixed scrub- *Simmondsia chinensis* |
| Tom's Thumb | Interior | Granitic Upland | Mixed scrub- *Simmondsia chinensis* |
| Tom's Thumb | Edge | Granitic Upland | Mixed scrub- *Simmondsia chinensis* |
| Gateway | Interior | Schist Hills | Mixed scrub- *Ambrosia deltoidea*, *Parkinsonia microphylla* |
| Gateway | Edge | Limy Upland | Mixed scrub- *Ambrosia deltoidea*, *Parkinsonia microphylla* |
| Sunrise | Interior | Limy Upland | Mixed scrub- *Ambrosia deltoidea*, *Parkinsonia microphylla*, *Larrea tridentata* |
| Sunrise | Edge | Clay Loam Upland | Mixed scrub- *Ambrosia deltoidea*, *Parkinsonia microphylla* |
| Prospector | Interior | Schist Hills | Mixed scrub- *Ambrosia deltoidea*, *Parkinsonia microphylla* |
| Prospector | Interior | Schist Hills | Mixed scrub- *Ambrosia deltoidea*, *Parkinsonia microphylla* |

these four interior-edge site pairs over eight years (2012–2020) to assess urban-edge effects. We also sampled one location (Prospector) with a similar configuration, but for which there was no development near that part of the Preserve boundary (i.e., both sites at this location are on the interior of the Preserve), for two years (2012–2014).

Sites shared similar climates (Fig 1) but differed slightly in elevation, ranging from 516m to 893m. Mean monthly temperatures during our study ranged from 10.0–35.0˚C at lowest elevation to 7.2–32.2˚C at the highest elevation (Fig 1B). Total monthly precipitation during our study ranged from 0–4.5cm at the lowest elevation to 0–6.7cm at the highest elevation (Fig 1C). Plant community associations were the same within paired sites, except two site pairs where interior sites overlapped a second plant community (Table 1). The most common community associations were *Ambrosia deltoides—Parkinsonia mycrophylla* and *Simmondsia chinensis* mixed scrub associations (Table 1), with spacing among plants consisting of mostly barren soil and rocks or invasive winter annual grasses (e.g. *Bromus rubens*, *Schismus barbatus*). Soil types were the same within interior-edge site pairs, except for two site pairs (Table 1). Soil types ranged in composition, dominated by clay loams, granites, schists, and limes.

## Ground-dwelling arthropod sampling

We sampled ground-dwelling arthropods according to protocols outlined by the Central Arizona–Phoenix Long-Term Ecological Research (CAP LTER) available in detail here: https://github.com/CAPLTER/caplter-research-protocols/tree/master/Arthropods. At each of the ten sites (five edge sites and five interior sites), we placed 10 traps at 5-m intervals along a transect perpendicular to the predominant slope. Transects were selected to include similar geomorphological characteristics, such as elevation (610–914m), slope (≤20%), and aspect (0–270˚, 315–360˚), to minimize extraneous factors. We positioned all transects such that they were concealed from view but near (<50m) existing trails or roads to facilitate access and limit off-trail travel. We constructed semi-permanent traps by burying 10cm of PVC pipe flush with the soil surface at each point along the transect. During sampling, we inserted a ~470mL transparent plastic cup into each pipe and left it exposed for ~72h after which we preserved the contents of the traps in ethanol then sent them to Arizona State University for processing. We covered traps when not in use. We treated empty traps as zero-samples in analyses. We lost some samples from flooding or collection difficulties, which are recorded in S2 Table. We sampled arthropods approximately quarterly for eight years (2012–2020) for a total of 29 discrete collections (see S1 Table for details), which we segregated by season: winter (December–

February), spring (March–May), summer (June–September), and fall (October–November). We stopped sampling during the summer season after 2016 on account of logistical constraints.

Pitfall trapping is a common method for sampling ground-dwelling arthropod communities with equal intensity among treatments [17]. However, ground-dwelling arthropods are notoriously difficult to identify. We sorted arthropods to morphologically similar groups, which were identified by CAP LTER personnel using a reference collection and appropriate keys. We used the common approach of identifying arthropods to the lowest practical taxonomic (LPT) level (*sensu* [18–21]). This method provides the highest taxonomic resolution reasonably possible allowing inferences into diversity patterns resulting in individuals assigned to a mixed-bag of taxa. We identified 22% of individuals to species, 13% to genus/subgenus, 43% to family/subfamily, 10% to order/suborder/superfamily, and 12% to class/subclass. From these LPTs, we chose seven distinct and abundant groups to examine abundance patterns separately: mites (subclass Acari), bristletails (order Microcoryphia), spiders (order Araneae), beetles (order Coleoptera), true bugs (order Hemiptera), ants (family Formicidae), and springtails (class Collembola).

## Climate data

We acquired precipitation and temperature measurements (Fig 1) from the Maricopa County Flood Control District (MCFCD), which operates a network of weather stations throughout central Arizona. For each arthropod sampling site, there was a MCFCD weather station featuring precipitation data within 4km and temperature data within 9km (S1 Table).

## Data analysis

To evaluate inventory completeness of each site (i.e. how close our sampling came to documenting all species present on site), we calculated sample coverage (i.e. an index of the completeness of field samples based on the number of detected rare species) using the function *iNext* and extrapolations of diversity using the function *estimateD* in the R package *iNEXT* [22]. We analyzed ground-dwelling arthropod alpha- and beta-diversity using treatment (interior vs edge), temperature (average during sampling period), and precipitation (total monthly precipitation for the 30 days prior to sampling) as predictors. We used R 3.6.2 (R Core Team 2023) for all analyses (R script and datasheets available in S1 File).

**Alpha diversity comparison between site pairs.** To compare alpha diversity of arthropods, we characterized and compared Hill numbers [23] for each site pair: (i) LPTs (i.e. richness, q = 0), (ii) exponential of Shannon's entropy index (i.e. Shannon's Diversity Index, q = 1), and (iii) inverse of Simpson's concentration index (i.e. Simpson's Diversity Index, q = 2). The higher order Hill numbers (q = 1 and q = 2) account for species evenness. In this analysis, we estimated the total Hill numbers for each site over the entire sampling period, then compared the interior and edge Hill numbers within each site pair. We extrapolated/ interpolated Hill number values to equal 2000 unique samples using asymptotic Chao1 estimators [23] via the function *estimateD* in the R package *iNEXT* [22]. We compared the extrapolated Hill number values between site pairs (interior/edge), denoting significant differences with non-overlap of 95% confidence intervals.

**Climate and site effects on alpha diversity.** We explored the effects of season (spring, summer, fall, and winter), treatment (interior/exterior) and climate using generalized linear mixed models (GLMM). GLMMs account for random effects and differences in sample sizes while testing multiple predictor variables. We choose two climate variables for predictor variables: precipitation and temperature. Because arthropod responses may be delayed after

precipitation, we used the sum of precipitation in the month prior to sampling [24]. We used the average temperature the month before sampling, as temperature represents physiological constraints for arthropods [25]. We checked collinearity and found no correlation between our two continuous variables of temperature and precipitation.

Our GLMMs used alpha diversity metrics of LPT richness and abundance (total and seven major arthropod groups separately: mites, bristletails, spiders, beetles, true bugs, ants, and springtails) across all sites and dates as response variables. To account for missing pitfall traps, we used a log-offset of trap number. As our response variables were count data, we ran each model with both Poisson and negative-binomial distributions, choosing the distribution with the lowest Akaike information criterion (AIC) and corroborated with AICc. We selected best-fit models with only significant predictors through backwards, stepwise elimination of saturated models. We applied post-hoc pairwise comparisons using estimated marginal means for significant categorical variables (i.e. treatment and/or season) to test group differences. We checked final model performance graphically via diagnostic plots and performed GLMMs in the R package *glmmTMB* [26].

### Beta diversity

For beta diversity, we assessed ground-dwelling arthropod community composition through multiple analyses. To determine paired edge-interior site differences in beta-diversity, we partitioned beta-diversity into incidence-based turnover ($\beta_{jtu}$) and nestedness ($\beta_{jne}$) components [27]. In this method, incidence-based β-diversity is decomposed by subtracting Simpson index of dissimilarity (accounting only for species turnover, ($\beta_{jtu}$)) from Jaccard index of dissimilarity (accounting for total ($\beta_{JAC}$)) yielding nestedness ($\beta_{jne}$). We report results of site-pairs, which always had the same sampling intensities, and partitioned beta-diversity using R-package *betapart* [27]. We also constructed similarity matrices with Bray-Curtis similarity coefficients to create ordinations of arthropod communities. We used the average arthropod abundance of the individual pitfall traps for each unique sample date at each site to account for missing traps. We visualized results via multi-dimensional scaling plots and tested satisfactory fit of ordination with goodness of fit and Shepard diagrams. We used permutations (n = 999) to test for significance of treatment (interior/edge), precipitation, and temperature variables fitted to ordinations. We used R package *vegan* [28] to construct ordinations and R package *ecodist* [29] to test environmental variable relationships.

### Results

In total, we captured and identified 25,477 arthropod individuals belonging to 287 LPTs over eight years at all sites (S2 Table). Sample coverage for LPTs was high, with more than 97% coverage for all sites (S1 File), indicating that our sampling effort was sufficient to capture the majority of the arthropod community present at each site. Of the LPT samples collected, 74 were unique to a single site, while 40 were found at all sites. Mites and ants were the most abundant arthropod groups captured (Fig 2A), followed by springtails, beetles, true bugs, bristletails, and spiders. Beetles had the highest number of unique LPTs, followed by spiders, true bugs, ants, and non-ant Hymenoptera, while mites had fewer LPTs, at least in part due to lower taxonomic resolution for this group. Across sampling periods, average arthropod abundance varied from approximately 4 to 31 individuals per sample, and the average number of LPTs varied from approximately 2.9 to 5.2 per sample (Fig 2C and 2D). Notable peaks in abundance were observed in the summer of 2012 (Fig 2C) with approximately 31 individuals per sample and in the summer of 2016 (Fig 2D) with approximately 5.2 LPTs per sample taxa.

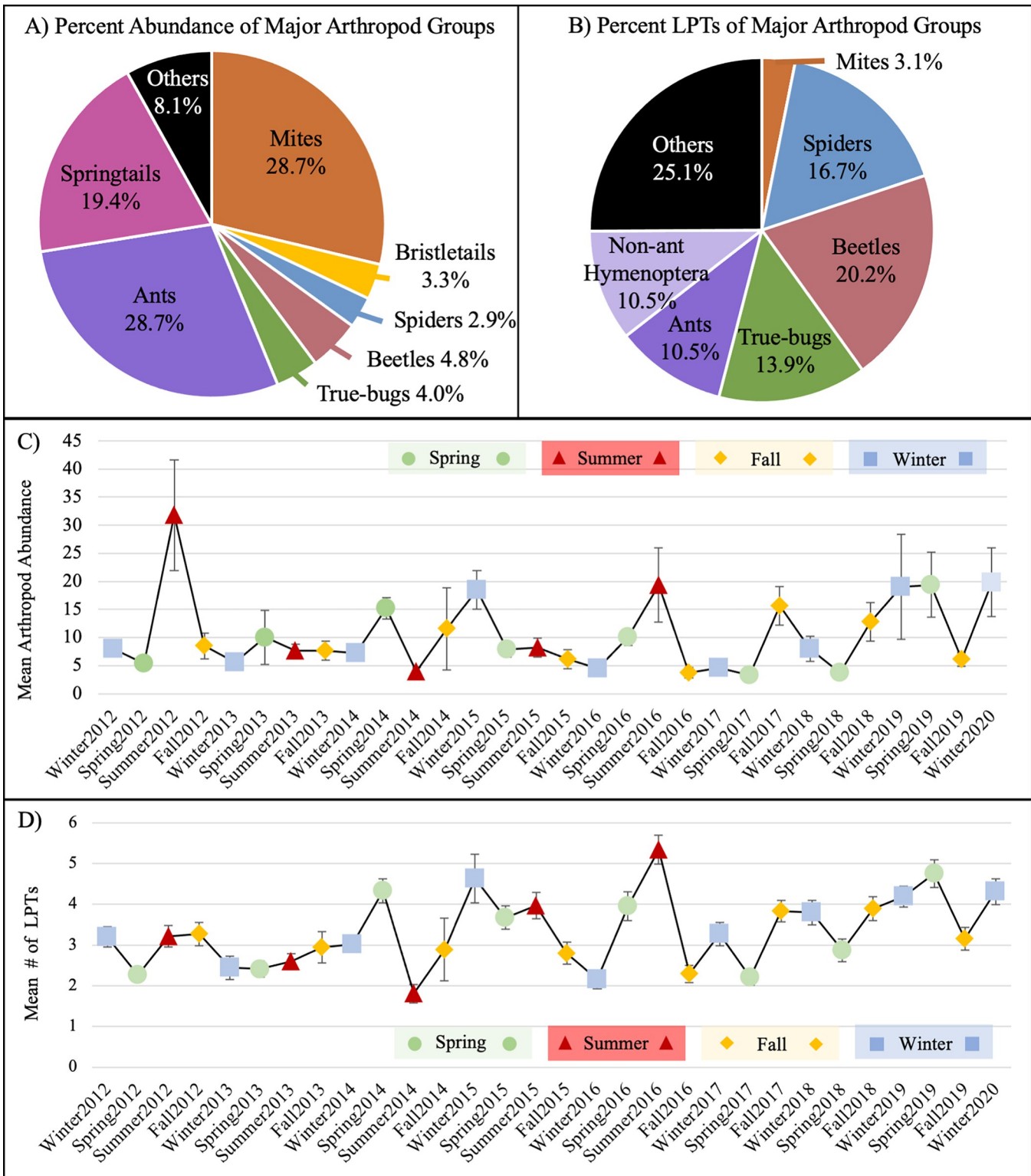

**Fig 2. Arthropod community characteristics.** Major arthropod groups percent by abundance (A) and LPTs (B), as well as average abundance (C) and average number of LPTs (D) per season/year, from eight years of sampling. Error bars show standard error.

## Alpha diversity: Arthropod richness and abundance differences between interior and edge sites

Two site pairs had significantly (i.e., non-overlap of 95% confidence intervals) larger numbers of LPTs (Hill number: q = 0) on the interior compared to the edge (Fig 3A). One interior site also had significantly higher 2nd (q = 1) and 3rd (q = 2) order Hill numbers, associated with evenness, compared to the edge site (Gateway, Fig 3A–3C). Contrary to hypothesis 1, one out of the site pairs had significantly higher order Hill numbers (q = 1 and q = 2) on the edge compared to the interior (Lone Mountain, Fig 3B and 3C). Additionally, the pair for which both sites were on the interior (Prospector) showed significant differences in higher order Hill numbers (q = 1, q = 2), showing sites differed in evenness on the interior regardless of urban edge (Fig 3B and 3C). One site pair (Sunrise) showed no significant differences at all.

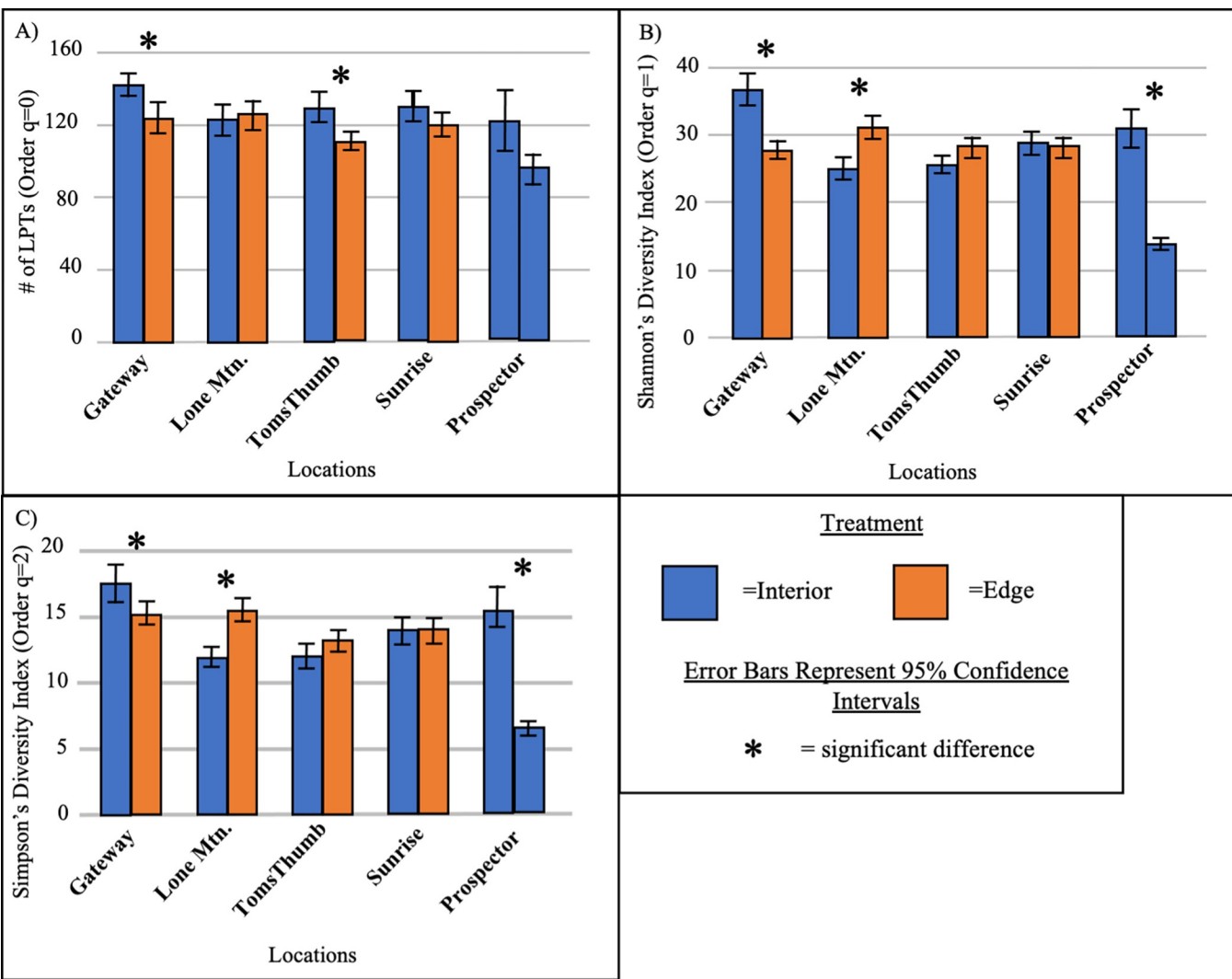

**Fig 3. Estimated site richness.** Hill number estimations of site pairs showing LPTs (i.e. richness, q = 0, panel A), exponential of Shannon's entropy index (q = 1, panel B), and inverse of Simpson's concentration index (q = 2, panel C). Bars denote one standard error of the mean. Asterisks denote significant differences (i.e. non-overlap of 95% confidence intervals) between pairs.

## Beta-diversity: Arthropod composition differences between interior and edge sites

The composition of arthropod communities varied between site pairs. A total of 192 LPTs shared between interior and edge sites, with 44 LPTs occurring exclusively in interior sites, and 48 LPTs occurring exclusively in edge sites. Some of these LPTs occurred at single sites, so may be just considered rare, rather than exclusive to edge or interior habitats. Therefore, we concentrate here on LPTs that occurred exclusively at two or more edge/interior sites (Table 2). Fifteen LPTs were exclusive to two or more edge sites. Bee flies (Bombyliidae) were exclusive to three edge sites, and one weevil genus (Curculionidae: *Ophryastes*) was exclusive to four edge sites (Table 2). Ten LPTs were exclusive to two or more interior sites (Table 2). Rover ants (Formicidae: *Brachymyrmex*) and goblin spiders (Oonopidae: *Scaphiella*) were both exclusive to three interior sites. Tarantula hawk wasps (Pompilidae: *Pepsis*) and army ants (Formicidae: *Neivamyrmex*) were exclusive to four and five interior sites, respectively (Table 2).

Overall, site compositional differences averaged 50.9% LPTs between sites, 39.7% of which was driven by turnover (i.e. one LPT replaced by another) and 10.6% by nestedness (i.e. loss of a LPT). The differences in LPT composition among interior and edge sites were relatively consistent, ranging between 51.6% to 56.9% shared species and 19.4% to 27.6% unique LPTs (Fig 4A). The paired interior control sites (Prospector, which were only sampled for two years) had

**Table 2. Arthropods in interior or exterior sites.** Arthropod LPTs exclusive to (>2) edge or interior sites. Lowest taxonomic resolution and number of occurrences at interior or edge sites are given.

| Class | Order | Family | Genus/species | Interior | Edge |
|---|---|---|---|---|---|
| Arachnida | Araneae | Dictynidae | NA | 0 | 2 |
| Arachnida | Araneae | Oonopidae | *Scaphiella* | 3 | 0 |
| Arachnida | Araneae | Thomisidae | NA | 2 | 0 |
| Chilopoda | Lithobiomorpha | Henicopidae | NA | 0 | 2 |
| Chilopoda | Scutigeromorpha | Scutigeridae | NA | 0 | 2 |
| Insecta | Coleoptera | Carabidae | *Selenophorus* | 0 | 2 |
| Insecta | Coleoptera | Chrysomelidae | *Chaetocnema* | 0 | 2 |
| Insecta | Coleoptera | Curculionidae | NA | 0 | 2 |
| Insecta | Coleoptera | Curculionidae | *Ophryastes* | 0 | 4 |
| Insecta | Coleoptera | Histeridae | *Euspilotus* | 0 | 2 |
| Insecta | Coleoptera | Tenebrionidae | *Eleodes longicollis* | 2 | 0 |
| Insecta | Coleoptera | Tenebrionidae | *Stenomorpha* | 0 | 2 |
| Insecta | Diptera | Bombyliidae | *Mythicomyia* | 0 | 2 |
| Insecta | Diptera | Bombyliidae | NA | 0 | 3 |
| Insecta | Hemiptera | Miridae | *Semium* | 2 | 0 |
| Insecta | Hemiptera | Reduviidae | *Apiomerus* | 2 | 0 |
| Insecta | Hemiptera | Rhopalidae | NA | 0 | 2 |
| Insecta | Hemiptera | Thyreocoridae | NA | 0 | 2 |
| Insecta | Hymenoptera | Diapriidae | NA | 0 | 2 |
| Insecta | Hymenoptera | Formicidae | *Brachymyrmex* | 3 | 0 |
| Insecta | Hymenoptera | Formicidae | *Neivamyrmex* | 4 | 0 |
| Insecta | Hymenoptera | Formicidae | *Nylanderia* | 2 | 0 |
| Insecta | Hymenoptera | Pompilidae | NA | 2 | 0 |
| Insecta | Hymenoptera | Pompilidae | *Pepsis* | 4 | 0 |
| Insecta | Hymenoptera | Pteromalidae | NA | 0 | 2 |

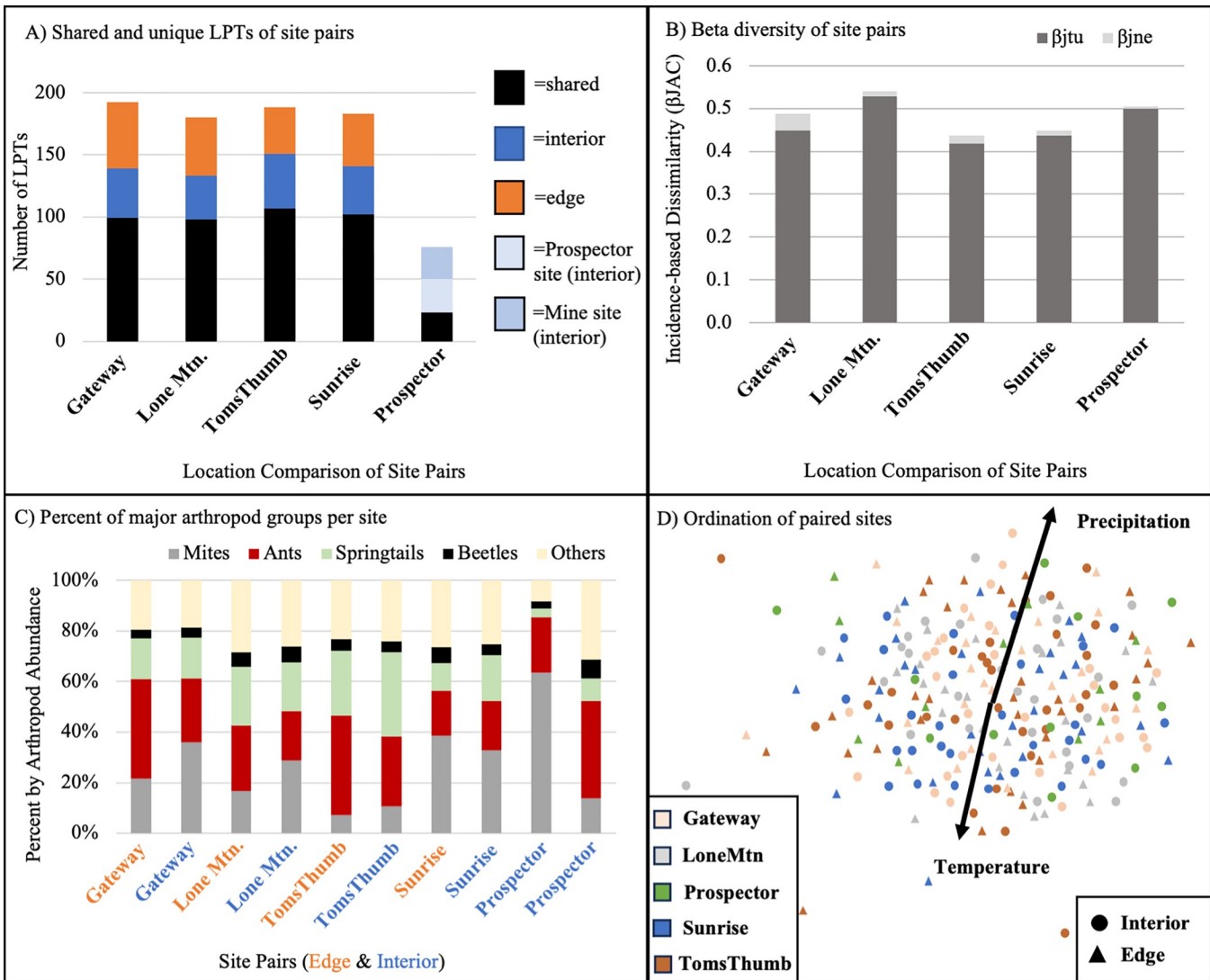

**Fig 4. Arthropod composition among sites.** Beta diversity dynamics between interior-edge treatments within each site in the McDowell Sonoran Preserve, Scottsdale, Arizona, USA with (A) number of shared/unique LPTs (note: the Prospector site pair was only sampled for two out of eight years), (B) dissimilarity analysis of ground-dwelling arthropods partitioned into incidence-based turnover ($\beta_{jtu}$) and nestedness ($\beta_{jne}$) of interior-edge site pairs, and (C) percent abundance of major arthropod groups per site and (D) multidimensional scaling plots of ground-dwelling arthropod communities in the McDowell Sonoran Preserve using Bray-Curtis similarity coefficient (stress = 0.307). Each point represents a unique site/date combination (i.e. average of all pitfall traps). Significant correlations of environmental variables are mapped onto the ordination (S1 File).

approximately one-third of LPTs shared and one-third of LPTs unique to either site. Site dissimilarities ranged from approximately 42% to 52%, driven mostly by turnover between site pairs (Fig 4B). At three out of four site pairs, edge sites had a higher arthropod abundance percentage of ants, while interior sites had more mites and springtails (Fig 4C). However, ordination of site compositions showed no significant differences between interior and edge sites or among sites (Fig 4D), suggesting a pool of abundant LPTs are shared among sites.

## Climate and arthropod community patterns

Precipitation was a significant predictor variable for multiple measures of arthropod alpha- and beta-diversity. Over eight years of sampling, total precipitation for the 30-days prior to the

**Table 3. Predictor variables for arthropod measurements.** Best generalized linear mixed models for ten arthropod response variables from backwards stepwise elimination of a saturated model (y = Treatment*Season + Precipitation + Temperature; full model output: S3 Table). Only significant (p<0.05) predictor variables are shown.

| Response | Best model |
|---|---|
| Arthropod abundance | y = Treatment*Season + Precipitation |
| Arthropod LPTs | y = Treatment*Season + Precipitation |
| Ant abundance | y = Treatment*Season + Temperature |
| Beetle abundance | y = Season |
| Bristletail abundance | y = Season |
| Mite abundance | y = Treatment*Season + Temperature |
| Spider abundance | y = Season + Precipitation |
| True bug abundance | y = Precipitation |
| Springtail abundance | y = Season |
| Fire ant abundance | y = Treatment + Temperature |

sampling date was a significant predictor variable for arthropod abundance, LPTs, spiders, and true bugs, while temperature during sampling was a significant predictor variable for ant and fire ant abundance in the GLMMs (Table 3, S3 Table). Precipitation and temperature were both significantly correlated with arthropod composition in the ordination (Fig 4D).

## Urban edge effects mediated by season

Many trends differentiating interior and edge sites varied by season. The interactive effects of treatment (interior/edge) and season were significant predictor variables in GLMMs for arthropod abundance, LPTs, ants, and mites (Table 3, S3 Table). Specifically, arthropod LPTs were found to be higher in interior sites compared to edge sites during the fall but were lower in the summer (Fig 5B). Arthropod abundance was greater in interior sites compared to edge sites during the spring but were lower in the summer (Fig 5A). These patterns were largely driven by ants and mites, which were higher in abundance in edge sites during summer and in interior sites during spring (Fig 5C and 5G). Fire ants were significantly higher in edge sites (Table 3), with their highest abundance observed in edge sites during summer (Fig 5F). However, we did not find significant differences in beetle, bristletail, true bug, spider, and springtail abundance between interior and edge sites (Fig 5D, 5E, 5H–5J), although these groups did differ by season and/or precipitation.

## Discussion

Overall, temperature and precipitation were strong drivers of arthropod trends, which are related to seasonal shifts that altered our observed patterns along the urban edge. We observed the urban edge effect changed seasonally, as many trends differentiating interior and edge sites varied by season. While interior sites had more arthropods in the spring, edge sites had higher numbers in the summer, largely driven by ants and mites, the two most abundant taxa. These findings suggest complex relationships among urbanization, climate, and arthropod communities in arid regions.

Our findings align with previous studies in the Sonoran Desert region, which have shown that urbanization can impact ground-dwelling arthropod communities ([12, 14, 15]). However, it is not always clear if these effects are detrimental or beneficial for arthropods. Typically, detrimental impacts to communities are measured by a reduction in biodiversity (e.g., abundance or richness) or change to community structure (e.g., dominance of nonnative organisms). We found two of four locations had significantly fewer LPTs (richness, q = 0) on the

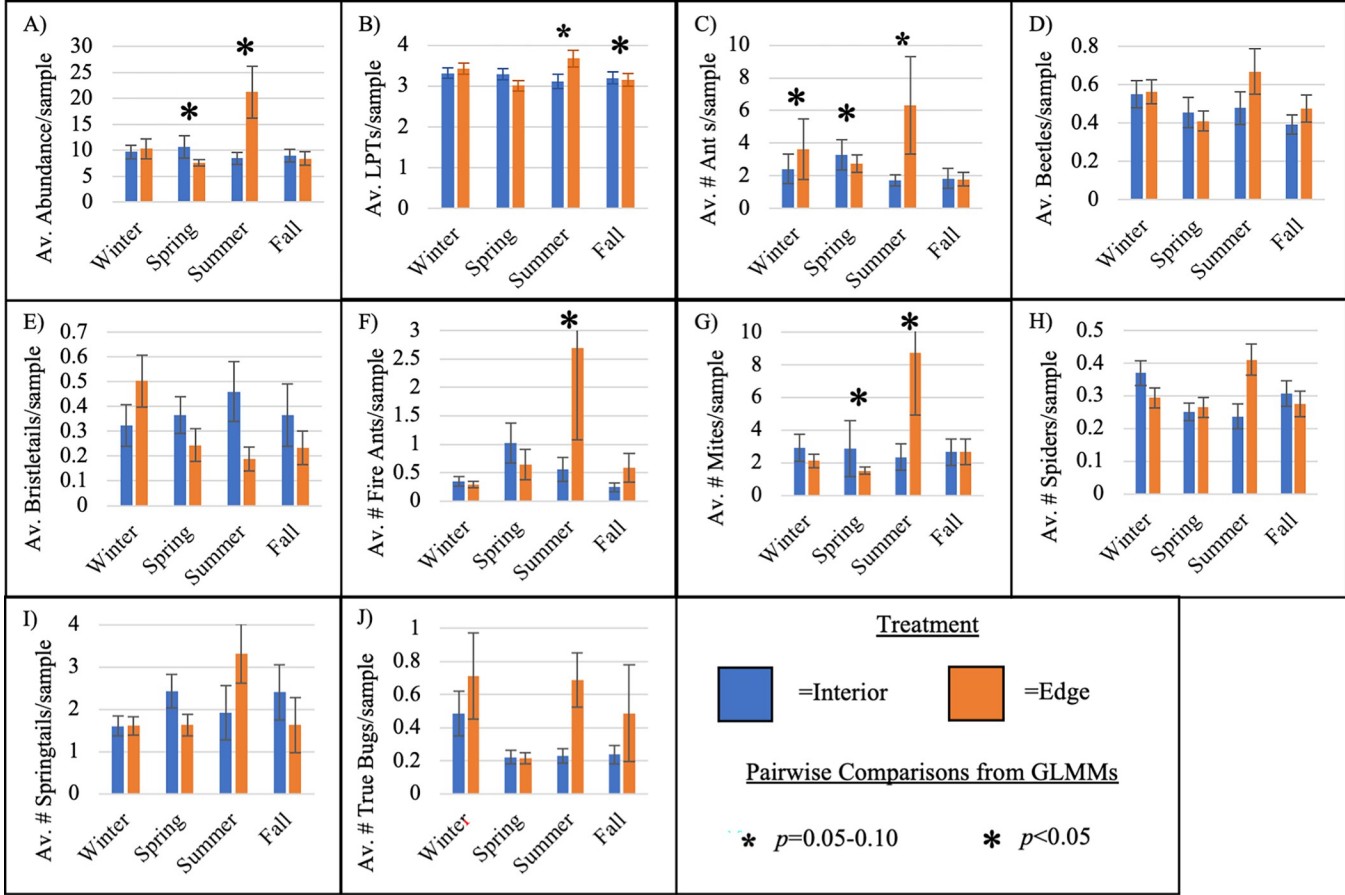

**Fig 5. Seasonal and site patterns of arthropods.** Average arthropod measures per season (winter, spring, summer, and fall) and treatment (interior/edge). Error bars denote standard error, while small asterisks show moderate significance ($p$ = 0.05–0.10) and large asterisks strong significance ($p<0.05$) derived from pairwise comparisons of generalized linear mixed models (see model output: S3 Table).

urban edge. However, the results for higher order Hill numbers ($q$ = 1 and $q$ = 2), which take into account evenness, were mixed. For example, at Lone Mountain, arthropods had higher diversity (second ($q$ = 1) and third order ($q$ = 2) Hill numbers), but not richness ($q$ = 0), near the urban edge than the interior. Furthermore, there were no significant differences at Sunrise for any Hill number. These results hint at possible, but not altogether clear, negative effects of the urban edge on arthropod communities.

Some of our patterns may be explained in part by soil and vegetation differences. Previous studies have suggested that the presence of unique vegetation in urban areas may promote higher arthropod diversity, possibly by providing habitat and food resources for a range of taxa [30]. The interior sites of Lone Mountain and Gateway were both at the transition of two plant communities and both had differences in Hill numbers. Lone Mountain was the only edge site with higher diversity values (Hill numbers) than its interior pair. At Gateway and Sunrise, soil types were different between the interior and edge sites. At Gateway, this may be an alternate explanation for the differences in Hill numbers, but at Sunrise, no differences in Hill numbers were detected. The Prospector sites (both interior) had shared soil type and plant community, but still had differences in Hill numbers. Thus, vegetation or soil differences do not fully explain arthropod differences and suggest more complex relationships among urbanization, climate, microsite, and arthropod communities in arid regions.

Sites were highly variable in species composition; between paired interior and edge sites the average dissimilarity was 50% of LPTs. This highlights the large diversity of the Sonoran Desert and the high variability in site conditions. Differences between interior and edge sites were driven by high turnover, the replacement of one LPT with another. Despite these differences, ordinations did not suggest significant compositional differences of treatment (interior/edge), season, and/or site. This may be due to lack of taxonomic resolution as ecological patterns often become more apparent with increased taxonomic resolution (e.g., [31, 32]). Therefore, our approach in using LTPs is conservative and may have underestimated differences. Another explanation may be extremely high variability in arthropod compositions of our system. Our findings underscore the complexity of arthropod communities across landscapes, where the urban edge is one of the many mechanisms that shape community dynamics.

## Seasonality and the urban edge

Seasonal changes may mediate an urban edge effects on arthropod communities. Specifically, interior sites had higher arthropod abundance and diversity during spring and fall, while edge sites had higher arthropod abundance and diversity during summer. The higher temperatures of summer particularly favored ants and mites on edge sites. Both of these groups are thermophilic and have many generalist strategies, which may be more adaptive in degraded habitats [33]. The coarse taxonomic resolution of mites in our study limits our ability to assess patterns that may be prevalent among mite subgroups, however we had high taxonomic resolution for ants (genera or lower).

We found ant increases during summer on the urban edge were largely driven by a single non-native and invasive species, fire ants. These patterns may be important to management of fire ants. For example, monitoring and control methods (e.g., pesticides, biocontrols, etc.) for fire ants may be most efficient during summer and along the urban edge. Fire ants are favored by the urban heat effect [34], which can increase temperatures over 10˚C in urbanized areas. Managers may seek to reduce urban heat (e.g., by planting trees, maintaining a natural buffer of undisturbed habitat, etc.) on the edges of natural preserves as a way to reduce fire ants.

We found support of our hypothesis that climate would be a strong driver of arthropods. There was a positive relationship between monthly precipitation and arthropod abundance, LPTs, spiders, and true bugs, while temperature had a positive relationship to ant abundance, similar to other studies in our area (e.g. [35–38]). The increased temperatures and added variation to precipitation patterns from climate change predicted for the Sonoran Desert is likely to challenge many native arthropod groups while favoring invasive species such as fire ants [34].

## Conclusion

Here we highlight that in arid regions with changing climates, preserve managers should consider how seasonal changes interact with urban edges to affect biodiversity. Shifts in community structure, such as invasive species cause, may be more likely to occur during certain seasons. Urban-wildland interfaces can act as species filters [39], but during certain seasons and under the pressure of climate change, those filters may break. Understanding these patterns can help preserve managers to best target locations and times to mitigate detrimental changes to biological communities in protected areas.

## Supporting information

**S1 Table. Site locations, characteristics, and weather stations.** Includes site coordinates, sampling dates, elevation, disturbance conditions, soil type, plant community, and weather

station proximity.
(XLSX)

**S2 Table. Arthropod taxa and site occupancy.** Includes all arthropod taxa found during sampling and which site they occurred at.
(XLSX)

**S3 Table. GLMM details.** Includes statistical details (e.g. z- and p-values) for GLMMs.
(XLSX)

**S1 File. R output and csv files.** Includes raw R-output along with csv files of raw data referenced in R-code. Arthropod data was obtained from [40].
(DOCX)

## Acknowledgments

We thank the many citizen scientists at the McDowell Sonoran Conservancy who have maintained and collected pitfall traps over the many years of this study, especially Steve Jones, Joanne Goldberg, and McDowell Sonoran Conservancy staff, Jessie Dwyer and Ayla Markowski for characterizing site disturbances. We thank the CAP LTER entomologists, who identified and enumerated all of the organisms. We thank Pam Templeman for organizing weather data. Our gratitude to the McDowell Sonoran Preserve and City of Scottsdale for site access.

## Author Contributions

**Conceptualization:** Stevan Earl, Tiffany Sprague, Helen Rowe.

**Data curation:** Stevan Earl, Jerry Holden, Helen Rowe.

**Formal analysis:** Derek A. Uhey, Richard W. Hofstetter, Stevan Earl, Helen Rowe.

**Investigation:** Stevan Earl, Jerry Holden, Tiffany Sprague, Helen Rowe.

**Methodology:** Stevan Earl, Jerry Holden, Tiffany Sprague.

**Project administration:** Stevan Earl, Jerry Holden, Tiffany Sprague, Helen Rowe.

**Software:** Derek A. Uhey.

**Supervision:** Stevan Earl, Jerry Holden, Helen Rowe.

**Visualization:** Derek A. Uhey.

**Writing – original draft:** Derek A. Uhey, Richard W. Hofstetter.

**Writing – review & editing:** Derek A. Uhey, Richard W. Hofstetter, Stevan Earl, Jerry Holden, Tiffany Sprague, Helen Rowe.

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
