## [Decision Letter · Decision Letter 0]

24 Oct 2023

PONE-D-23-29401Living on the edge: The sensitivity of arthropods to development and climate along an urban-wildland interface in the Sonoran Desert of central ArizonaPLOS ONE

Dear Dr. Uhey,

Thank you for submitting your manuscript to PLOS ONE. After careful consideration, we feel that it has merit but does not fully meet PLOS ONE’s publication criteria as it currently stands. Therefore, we invite you to submit a revised version of the manuscript that addresses the points raised during the review process.

We look forward to receiving your revised manuscript.

Kind regards,

Ramzi Mansour

Academic Editor

PLOS ONE

Journal Requirements:

6. We note that Figure 1 in your submission contain [map/satellite] images which may be copyrighted. All PLOS content is published under the Creative Commons Attribution License (CC BY 4.0), which means that the manuscript, images, and Supporting Information files will be freely available online, and any third party is permitted to access, download, copy, distribute, and use these materials in any way, even commercially, with proper attribution. For these reasons, we cannot publish previously copyrighted maps or satellite images created using proprietary data, such as Google software (Google Maps, Street View, and Earth). For more information, see our copyright guidelines: http://journals.plos.org/plosone/s/licenses-and-copyright.

Reviewers' comments:

Reviewer's Responses to Questions

**Comments to the Author**

1. Is the manuscript technically sound, and do the data support the conclusions?

Reviewer #1: Yes

Reviewer #2: Yes

2. Has the statistical analysis been performed appropriately and rigorously? 

Reviewer #1: N/A

Reviewer #2: Yes

3. Have the authors made all data underlying the findings in their manuscript fully available?

Reviewer #1: Yes

Reviewer #2: Yes

4. Is the manuscript presented in an intelligible fashion and written in standard English?

Reviewer #1: Yes

Reviewer #2: Yes

5. Review Comments to the Author

Reviewer #1: In this study, the authors compared ground-dwelling arthropod community composition between edge and interior sites of a Preserve partially surrounded by urban areas. I believe the collected data is valuable, and the question regarding whether the effects of urbanization on arthropod communities spill over into adjacent natural areas is highly relevant. Nevertheless, I think that the study’s focus should be more explicit, and the hypotheses/expectations should be better justified. All my comments are detailed in the attachment.

Reviewer #2: Lines 49-52: Authors reference that arthropods in arid regions adapt to hot and dry environments but do poorly in urban areas because of these adaptations. The authors need to add a sentence of justification for this since cities are often hotter and dryer than rural areas. Perhaps that these are already extreme environments and any additional perturbation makes them vulnerable. As it states now, I am not convinced.

Lines 124-129: Was there anything at the bottom of the cup in the pitfall trap to keep insects from crawling or jumping out of the trap? Were they set out as dry? This will affect the types and counts of arthropods and while they would be consistent across sampling, it does bias some of the end results.

Lines 185-195: Did you also test for year? The samples were collected across multiple years and the methods do not state how the year of collection was handled. Were the seasons averaged across years or individual per year?

Table 2 would be helpful if there was an additional column that included native/invasive. Were the edge exclusive LPTs because they were non-natives associated with urban or were they species that typically do well in edge regions?

6. PLOS authors have the option to publish the peer review history of their article (what does this mean?). If published, this will include your full peer review and any attached files.

Reviewer #1: No

Reviewer #2: No

---

## [Author Response · Author response to Decision Letter 0]

28 Dec 2023

Please see attached document, "Response to Reviewers"

---

## [Editor Report · Decision Letter 1]

2 Jan 2024

Living on the edge: The sensitivity of arthropods to development and climate along an urban-wildland interface in the Sonoran Desert of central Arizona

PONE-D-23-29401R1

Dear Dr. Uhey,

We’re pleased to inform you that your manuscript has been judged scientifically suitable for publication and will be formally accepted for publication once it meets all outstanding technical requirements.

Kind regards,

Ramzi Mansour

Academic Editor

PLOS ONE
---

## [Editor Report · Acceptance letter]

8 Apr 2024

PONE-D-23-29401R1 

PLOS ONE

Dear Dr. Uhey, 

I'm pleased to inform you that your manuscript has been deemed suitable for publication in PLOS ONE. Congratulations! Your manuscript is now being handed over to our production team.

Kind regards, 

on behalf of

Dr. Ramzi Mansour 

Academic Editor

PLOS ONE